# Exploring the Spatiotemporal Characteristics and Causes of Rear-End Collisions on Urban Roadways

**Wenhui Zhang, Tuo Liu * and Jing Yi**

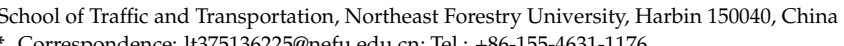

School of Traffic and Transportation, Northeast Forestry University, Harbin 150040, China
*   Correspondence: lt375136225@nefu.edu.cn; Tel.: +86-155-4631-1176

**Abstract:** Rear-end collisions are caused by drivers misjudging urgent risks while following vehicles ahead in most cases. However, compared with other accident types, rear-end collisions have higher preventability. This study aims to reveal the prone segments and hours of rear-end collisions. First, we extracted 1236 cases from traffic accident records in Harbin from 2015 to 2019. These accidents are classified as property damage accidents, injury accidents and fatal accidents according to the collision severity. Second, density analysis in GIS was used to demonstrate the spatial distribution of rear-end collisions. The collision spots considering the density and severity were visually displayed. We counted the hourly and seasonal distribution characteristics according to the statistical data. Finally, LightGBM and random forest classifier models were used to evaluate the substantial factors affecting accident severity. The results have potential practical value in rear-end collision warning and prevention.

**Keywords:** rear-end collision; spatiotemporal analysis; machine learning model; collision prediction; urban traffic analysis

## 1. Introduction

With the rapid development of the economy and society, the urbanizing process is accelerating each year. The urban road network and the relevant infrastructure have been gradually completed. The increasing proportion of motorized travel poses challenges for traffic safety. The frequent collision types are also becoming different from those in the past. In many countries, rear-end collisions are considered to be the most frequent type of total traffic accidents [1]. Statistics show that rear-end collisions account for approximately 30% of all crashes and 20% of all fatal traffic accidents. In China, this proportion is even more severe. For a higher relative crash speed, rear-end collisions tend to result in more severe property damage and personal injury. It has been proven that developing and applying active safe equipment in vehicles is an effective method to prevent traffic accidents. However, identifying hazard segments and improving traffic safety facilities are also important issues.

Different traffic environments, driving performances and involved vehicle types result in different degrees of severity [2]. A number of studies have focused on deep analysis of the rear-end crash course, aiming to find the prominent factors influencing the collision occurrences. According to the results, a series of prevention mechanisms can be proposed to improve the safety level. In addition, some studies propose spatial autocorrelation methods to analyze the potential spatiotemporal patterns of collisions. Considering the multiple factors in the model, it can also assess the consequences of the accidents [3]. Combining information on traffic flow, traffic rules, road geometry and human factors, Bayesian spatiotemporal models [4] and probabilistic models [5] provide a good description of collision occurrence trends. The generalized linear mixed model [6], gravity model [7] and regression model [8] yield good results in the analysis of factors influencing rear-end collision severity. The factors and characteristics of rear-end collisions derived from the

study can provide support for prediction. In recent years, mathematical statistics and machine learning have been widely used for accident data mining. Neural networks [9], stochastic parametric models and negative binomial models [10] are popular for risk assessment and prediction of rear-end collisions.

This paper aims to analyze the spatial and temporal distribution characteristics of rear-end collisions based on the statistical data of rear-end collisions in Harbin. Two machine learning algorithms are used to predict the severity of rear-end collisions and identify the key factors that influence the severity.

## 2. Literature Review

Many studies have focused on the characterization and causation of traffic accidents. Historical accident data are used to explore the important causes of traffic accidents. Furthermore, geographic information systems (GISs) are generally employed to analyze the accidents' spatial and temporal distribution [11]. Some studies have shown the possible correlation between injury severity and segments that occur [12]. As a classical posterior probability model, the Bayesian model is widely used to reveal crash characteristics [13]. Additionally, the Spider Spot and the Kernel Density Estimation (KDE) method has been proven to be effective for analyzing the temporal and spatial patterns of collisions [14].

In addition to collision distribution, some studies concentrated on the influence of traffic accident frequency using spatiotemporal analysis. The current study found that the time and space parameters have a large effect on the collision frequency [15]. Spatiotemporal models allow the study to obtain the regional traffic accident frequency trends [16]. It was concluded that spatial and temporal factors were found to be significantly correlated in accident frequency [17]. This has important implications for the level of dispersion of crash data [12]. In addition, spatial effects play a more important role than temporal effects, but temporal factors are still essential in spatiotemporal models.

With the increase in computing power and the development of big data, machine learning has been introduced into traffic accident research. Machine learning has powerful feature extraction capabilities. Moreover, machine learning also holds great prospects in the field of traffic accident research [18]. Incorporating traffic information in machine learning models can quickly obtain the traffic characteristics of a region or a city [19]. Modeling using these traffic characteristics can predict the traffic and accident risk in this area [20]. Machine learning is effectively applied to traffic accident research based on vehicle trajectory data [21]. This method can generate simulated trajectory data [22]. Accident types can be identified by analyzing a large number of vehicle trajectory trends [23].

Machine learning is also applied to predicting traffic accidents. Established prediction models often have errors in fatal and injury collisions [24]. To address this disadvantage, some scholars have introduced deep learning into the model of accidental collisions by defining loss functions to predict the accident severity. A CNN-based deep learning algorithm can increase the optimization of model accuracy [25].

Rear-end collisions have become a special topic of study for traffic accidents because they have a high frequency. The main research focus is on the causes of the accidents. In the existing studies, the influencing factors of rear-end collisions are generally investigated by driving simulation test analysis [26] and constructing simulated crash models [27].

Many studies have focused on the influence of human factor in crashes. In some researches, simulation experiments of driving scenarios are generally used to collect data on the driving behavior characteristics of drivers. The data are analyzed by algorithms and models to derive the effects of different human factors on crashes. Driver's risk reaction time [28], speed and surroundings perception [29], sleep deprivation [30], driver's experience [31] and driver's intention [32] are often used as research objects.

In other research, crash data sources are mainly accident records that have already occurred. Crash analysis models are developed on this to analyze crash factors. Descriptive statistical analysis [33] and binary logit models [34] are often used to study the causal factors of accidents. The causes of rear-end accidents are complex and involve a large number of



variables [35]. Machine learning models have an advantage in dealing with these types of problems. Theofilatos and Yannis [36] used a random forest model and a Bayesian logistic regression model to reveal the most important accident variables. Wang et al. [37] constructed a rear-end collision prediction mechanism (RCPM) based on deep learning methods.

Machine learning is also very efficient when combined with other algorithms for modeling and solving problems. The GA-XGBoost feature recognition model can accurately identify urban traffic accident features [38]. The model can quickly extract factors including driving experience, illegal driving, road intersection types, weather, traffic flow and time intervals [39]. The rear-end collision model is one kind of traffic accident risk model. Neural network is commonly used as a research method in the study. Hybrid neural network models and deep learning models have good performance in complex feature extraction of traffic safety accidents [40].

In summary, the existing studies on rear-end collisions are relatively microscopic. Most consider modeling analysis in specific scenarios. This paper is devoted to analyzing the spatial and temporal distribution characteristics of regional rear-end collisions and considering the prominent seasonal characteristics of the cold in Harbin. A comparison between two machine learning models, LightGBM and Random forest, is also performed for prediction and causation analysis.

## 3. Methods

### 3.1. Mean Center

The mean center (MC) indicates the average location of the observed sample. It reflects the concentrating trend and the overall offset of the sample data in space. We can use the rear-end collision data to calculate MC. The concentration and offset trend can be obtained by calculating MC as follows:

$$\overline{X} = \frac{\sum_{i=1}^{N} x_i}{N} \tag{1}$$

$$\overline{Y} = \frac{\sum_{i=1}^{N} y_i}{N} \tag{2}$$

$\overline{X}$ and $\overline{Y}$ represent the average coordinates of data in the $X$ and $Y$ directions. $N$ is the total number. $x_i$ and $y_i$ represent the $X$ and $Y$ coordinates of the ith data point.

### 3.2. Standard Deviational Ellipse

The standard deviational ellipse (SDE) can be used to characterize the spatial distribution of data, including central, dispersion and directional distribution tendencies. The length of the short axis indicates the degree of spatial aggregation. The shorter the axis shows, the more aggregated the data are. The long axis represents the spatial expansion direction as follows:

$$\theta = arctan \frac{\left(\sum a_i^2 + \sum b_i^2\right) + \left\{\sum a_i^2 - \sum b_i^2 + 4\left(\sum a_i \sum b_i\right)^2\right\}^{\frac{1}{2}}}{2\sum a_i \sum b_i} \tag{3}$$

$$S_x = \sqrt{\frac{\sum (a_i cos\theta - b_i sin\theta)^2}{N - 2}} \tag{4}$$

$$S_y = \sqrt{\frac{\sum (a_i sin\theta - b_i cos\theta)^2}{N - 2}} \tag{5}$$

$a_i = x_i - \overline{X}$, $b_i = y_i - \overline{Y}$ and $\left(\overline{X}, \overline{Y}\right)$ are the mean center coordinates of the data, which can be calculated from Equations (1) and (2). $\theta$ is the angle of clockwise rotation of the standard deviation ellipse along the $Y$-axis. $S_x$ and $S_y$ represent the long and short semiaxes of the standard deviation ellipse, respectively. In this study, the standard deviation ellipse is obtained from the two-dimensional coordinates of all data points. It can evaluate the aggregation degree and expansion direction of data.

### 3.3. Density Analysis

The urban map can be divided into some small cells with a side length of *d*, which eventually corresponds to pixel cells on the GIS. *k* represents the center of the circle. *r* represents the radius. We can use $N_k(r)$ to calculate the number of events around the neighborhood. $N_k(r)$ is divided by the neighborhood area to obtain the accident density named $D_k{}^{accident}$. Similarly, we can obtain the density of the road network, $D_k{}^{road}$ as follows:

$$D_k{}^{accident} = \frac{N_k(r)}{\pi r^2} \tag{6}$$

$$D_k{}^{road} = \frac{L_k(r)}{\pi r^2} \tag{7}$$

If $q_i$ represents the severity of the *i*th accident, the density of accident severity named by $D_k{}^{severity}$ in cell *k* can be obtained as follows:

$$D_k^{severity} = \frac{\sum_{i=1}^{N_k(r)} q_i}{\pi r^2} \tag{8}$$

### 3.4. Clustering Analysis

The outlier analysis can be calculated by the local Moran's I of the data points.

$$I_i = \frac{m_i - \overline{M}}{S_i{}^2} \sum_{j=1, j \neq i}^{N} \omega_{i,j}(m_j - \overline{M}) \tag{9}$$

$I_i$ is the local Moran's I of point *i*. $m_i$ and $m_j$ are the attributes of points *i* and *j*. $\overline{M}$ is the global mean of the attribute. $\omega_{i,j}$ is the spatial weight between point *i* and point *j*. $\omega_{i,j}$ is usually the inverse of the distance between the two points. $S_i^2$ is the second-order sample matrix of all attributes except point *i*.

$$S_i^2 = \frac{\sum_{j=1, j \neq i}^{N} (m_j - \overline{M})^2}{N-1} \tag{10}$$

$$Z_{I_i} = \frac{I_i - E[I_i]}{\sqrt{V[I_i]}} \tag{11}$$

$$E[I_i] = \frac{\sum_{j=1, j \neq i}^{N} \omega_{i,j}}{N-1} \tag{12}$$

$$V[I_i] = E\left[I_i^2\right] - E[I_i]^2 \tag{13}$$

Normally, the confidence level of statistical significance is 95%. According to the normal distribution, the range of *z* should be between $-1.96$ and $+1.96$. In terms of statistical significance, if $I > 0$, this point has the same attribute level as the neighboring points. It is reflected as high–high clustering or low–low clustering. The result depends on the difference between the attribute value of this point and the average value of all points. If $I < 0$, there is a large difference in attributes between this point and the neighboring points. This point is an outlier.

There were five types of results: high–high clustering (H–H), high–low clustering (H–L), low–high clustering (L–H), low–low clustering (L–L) and nonsignificant. High–high clustering means that this point and its neighbors are all high values. They also have similar attributes. High–low clustering means that this point is a high value and is surrounded by neighboring points of low values. Low–high clustering means that this point is a low value and is surrounded by neighboring points of high values. Low–low clustering means that this point is low value and the neighboring points are also low value. Nonsignificant indicates that the point has no significant relationship with the neighboring points and the attributes are more different. In general, the local Moran's I describes the clustering characteristics between high and low values points in geo-spatial.

### 3.5. LightGBM

LightGBM is an efficient integrated machine learning algorithm developed by the Microsoft team based on the Boosting method. The purpose of the LightGBM algorithm is to find an approximation of a function. This function can minimize the specified loss function. The loss function will determine how well the model fits the data. The LightGBM model integrates multiple regression trees to fit the final model. First the algorithm needs to determine how the objective function is calculated. After that, the optimization problem is to make each tree have the smallest objective function. For this, it is necessary to calculate the gain from the splitting of the leaf nodes in the tree. When the maximum gain of node splitting is got, the feature with the highest gain will be selected as the splitting feature. Continuously this iterating process until a specific condition is met. LightGBM uses histogram algorithms, Leaf-wise growth strategies, and histogram differential acceleration and other methods. It can significantly reduce the complexity of the algorithm and training time consumption. This makes LightGBM has excellent training efficiency and high prediction accuracy.

### 3.6. Random Forest

Random forest (RF) is a supervised data mining algorithm. It is a combination of bagging algorithm and decision tree, which is an integrated algorithm. Its main workflow is as follows:

(1) RF uses the bagging algorithm to sample the training set with put back Bootstrap sampling. It forms sub-training set. In other words, each sub-training set is drawn from the original training set by put-back bootstrap sampling.

(2) Using the decision tree method, a binary tree corresponding to each sub-training set is formed.

(3) The algorithm repeats steps 1 and 2. When the generated tree can accurately classify the samples in the training set, the algorithm will come up with a tree model. Or, until all the attribute features are used up, the algorithm will generate a tree model. After that, the tree model of all processes is combined. This forms a random forest model.

(4) For an arbitrary test sample, the final classification result is decided by simple voting.

Random forests have two important randomness. One of them is the sample randomness of Bootstrap sampling, and the other is the random selection of features. Both of two randomness make it possible to greatly reduce the correlation of each tree in the random forest. Thus, it ensures that the random forest has good classification ability.

## 4. Data Analysis

### 4.1. Data Source

The data come from the Harbin Traffic Safety Management Database. The selected data are from 2015 to 2017 with a total of 8037 cases. The data attributes can be divided into five categories. The fields contained in each category are shown in Table 1. The rear-end collision data were filtered from the database for a total of 1372 pieces. Figure 1 shows the road network of Harbin.

**Table 1.** Categories and fields of traffic accident data records.

| Categories | Fields |
| --- | --- |
| Basic information | Number, district, date, time, location, etc. |
| Reason | Type, vehicle information, preliminary reason, determination reason, etc. |
| Evaluation | Participants, property damage, number of injuries, number of death, etc. |
| External environment | Weather, temperature, terrain, road physical separation, roadside protection facilities, road conditions, etc. |
| Other information | Road safety attributes, jurisdiction units, road safety supervision level, etc. |

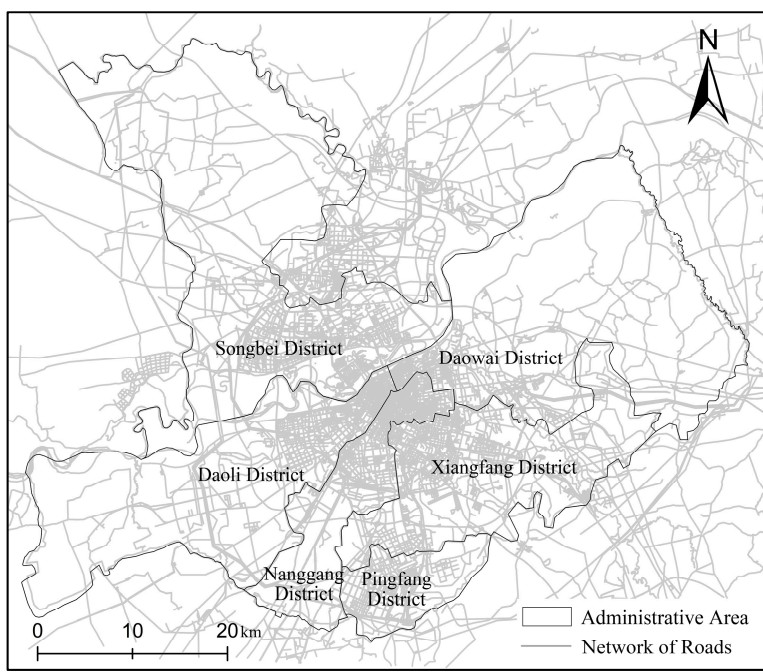

**Figure 1.** Road network of Harbin.

*4.2. Data Mining*

In the data processing, 136 pieces of data are missing some field in the "reason" categories; we deleted this data. A total of 264 pieces of data are missing or incorrect in the "weather" field; we corrected the details of these data by querying the weather records.

After data preprocessing, 1236 pieces of data completed the accident information. We selected the central city of Harbin as the study area. Within this area, a final 1205 samples were screened.

(1) According to the traffic accident severity classification, accidents are classified into three categories: property damage accidents, injury accidents, and fatal accidents. Accident severity can be a label for data, as shown in Table 2.

**Table 2.** Classification and standards of accident severity.

| Severity Level | Category | Standard |
|---|---|---|
| Level 1 | Property damage accidents | No casualties in the accident |
| Level 2 | Injury accidents | No fatalities in the accident, but some injuries |
| Level 3 | Fatal accidents | Accidents with fatalities |

The data were graded according to the level of severity, as shown in Table 3.

**Table 3.** Level of traffic accident severity.

| Level | Description | Frequency | Proportion |
|---|---|---|---|
| 1 | Property damage accidents | 616 | 51.12% |
| 2 | Injury accidents | 449 | 37.26% |
| 3 | Fatal accidents | 140 | 11.61% |

(2) In this paper, feature variables were extracted by mining collision data. Nine feature variables were extracted: weather, wind speed, temperature, week, season, time, location, vehicle type, and accident type. The feature variables were coded, as shown in Table 4.

**Table 4.** Descriptive statistics of characteristic variables of accident data.

| Feature Variables | Feature Description | Frequency | Proportion (%) |
|---|---|---|---|
| Season | Spring = 1 | 231 | 19.17% |
| | Summer = 2 | 411 | 34.11% |
| | Autumn = 3 | 328 | 27.21% |
| | Winter = 4 | 235 | 19.50% |
| Week | Weekdays = 0 | 903 | 74.94% |
| | Weekend = 1 | 302 | 25.06% |
| Wind speed | Below level 3 = 1 | 407 | 33.78% |
| | 3~4 levels = 2 | 532 | 44.15% |
| | 4~5 levels = 3 | 224 | 18.59% |
| | 5~6 level = 4 | 42 | 3.48% |
| Weather | Sunny day = 1 | 411 | 34.11% |
| | Cloudy day = 2 | 440 | 36.51% |
| | Snowy, rainy, foggy day = 3 | 354 | 29.38% |
| Time | Nighttime nonpeak hours (19:00~07:00) = 1 | 525 | 43.57% |
| | Daytime nonpeak hours (9:00~19:00) = 2 | 416 | 34.52% |
| | Peak hours (7:00~9:00 and 17:00~19:00) = 3 | 264 | 21.91% |
| Location | Road = 0 | 841 | 69.79% |
| | Intersection = 1 | 364 | 30.21% |
| Accident type | Three or more cars = 0 | 71 | 5.89% |
| | Two-car accident = 1 | 1134 | 94.11% |
| Vehicle type | Involving bus = 1 | 243 | 20.17% |
| | Involving trucks = 2 | 364 | 30.21% |
| | Small car = 3 | 567 | 47.05% |
| | Others = 4 | 31 | 2.57% |
| Temperature | Continuous Variables | 1205 | 100% |

(3) This step creates the coordinates' visualization for the collision location by GIS. After coordinate transformation, 1127 pieces of data were finally retained. For the follow study, we chose these 1127 samples with accurate coordinate conversion. Figure 2 shows the spatial distribution of collisions after data processing. The blue points represent collision locations.

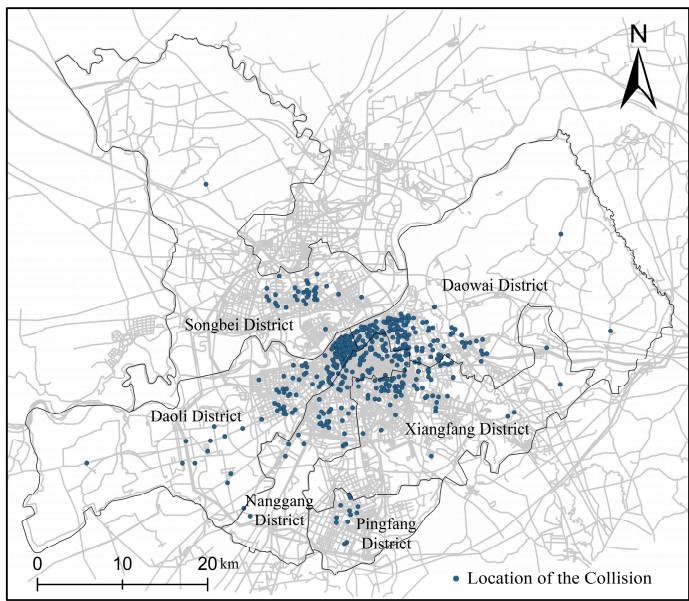

**Figure 2.** Spatial distribution of rear-end collisions.

### 4.3. Feature Extraction

We used the Pearson correlation coefficient to extract the features (Figure 3). The Pearson correlation coefficient $r$ utilizes values in the range of $(-1, 1)$. When $r \geq 0.8$, it can be regarded as highly correlated. When $0.5 \leq r < 0.8$, it can be regarded as moderately correlated. When $r < 0.3$, it means that the correlation between them is extremely weak. The matrix shows that except for temperature and season, which are moderately correlated, the correlation between each feature is weak. The nine feature variables can be used for analysis.

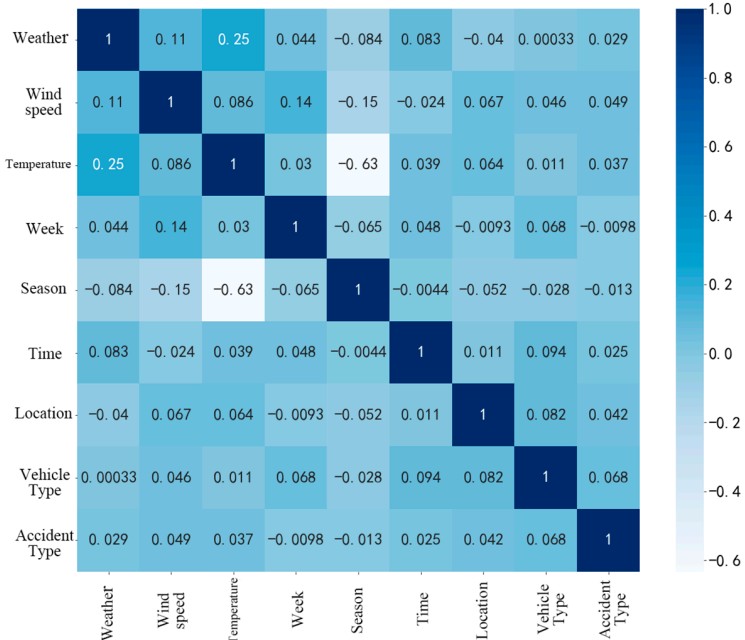

**Figure 3.** Correlation matrix.

## 5. Results

### 5.1. Spatial Distribution Characteristics

#### 5.1.1. Spatial Distribution

Applying Equations (1)–(5), the mean center location and ellipse distribution location of collisions are obtained. The spatial distribution trend is shown in Figure 4.

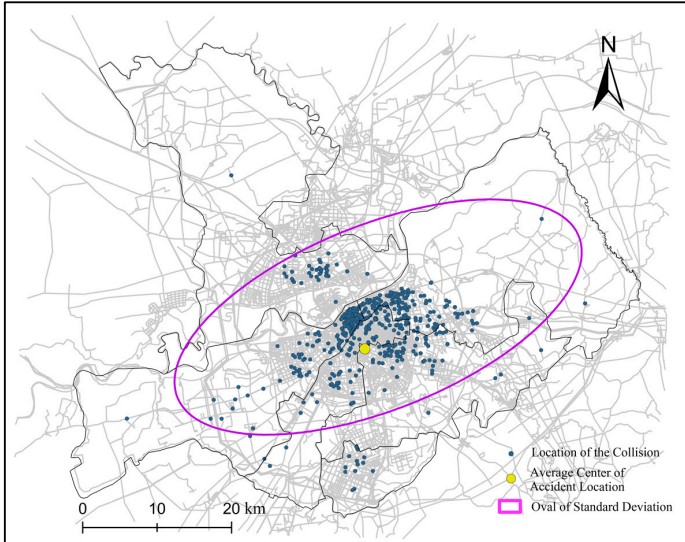

**Figure 4.** Spatial distribution direction.

As shown in Figure 4, most collisions are concentrated in the interface of the two administrative districts of Daoli and Daowai. The mean center is located south of the area.

The length of short axis of the standard deviation ellipse is quite short. It means that the spatial distribution is more concentrated. The direction of the long axis shows that the distribution has a trend of spreading from southwest to northeast.

### 5.1.2. Density Distribution

Equation (6) can obtain the density of collision points. The maximum normalization was used to process density values. Based on the density distribution map, 0.2, 0.5 and 0.8 are defined as dividing lines; 0~0.2 are defined as a low-density region. This means that there is a very small number of collisions per unit area; 0.2~0.5 is the medium–low region; 0.5~0.8 is the medium-high density region; 0.8~1 is the high-density region. The density calculations are shown in Figure 5.

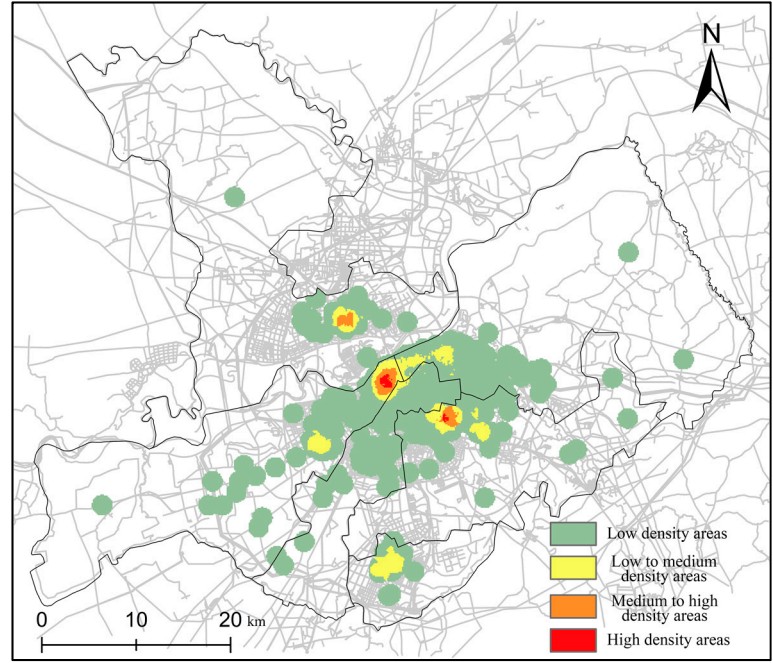

**Figure 5.** The density distribution.

In Figure 5, the low-density areas account for the largest proportion. The high-density areas account for the smallest proportion. Density region statistics are shown in Table 5.

**Table 5.** Distribution density region statistics.

|  | Low to Medium | Medium to High | High |
| --- | --- | --- | --- |
| Administrative District | Daoli, Daowai, Xiangfang, Pingfang, Songbei | Daoli, Xiangfang, Songbei | Daoli, Xiangfang |
| Area(km$^2$) | 26.7 | 7.3 | 1.3 |

In this research, we reviewed the 2020 annual road network density monitoring report for major Chinese cities. The report shows that there is a crucial difference in road network density in different districts of Harbin. The maximum value is 6.4 km/km$^2$ in Daoli, the minimum is 3.6 km/km$^2$ in Pingfang and the mean value is 5.0 km/km$^2$. To reduce the influence on the results, this paper obtains the accident frequency per unit of road length. The results are shown in Figure 6.

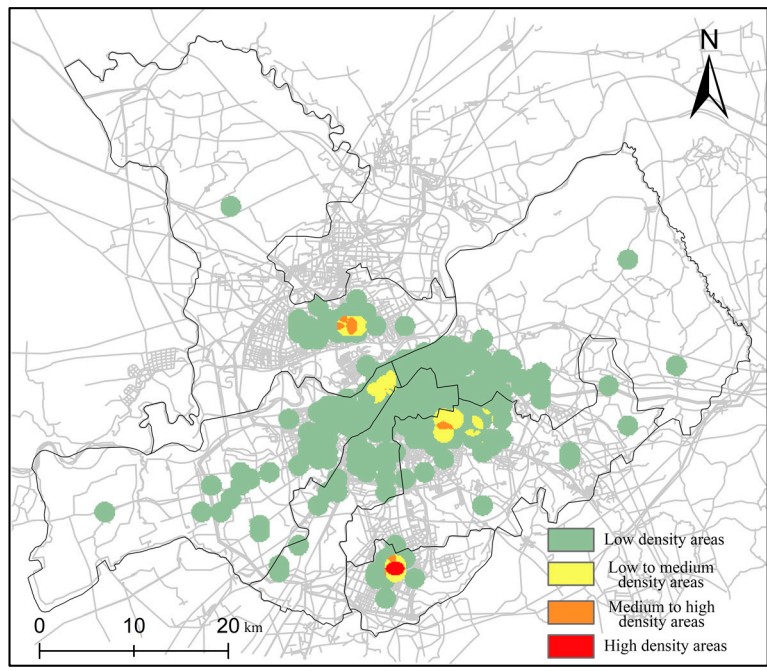

**Figure 6.** The density distribution considering the road network density.

In Figure 6, the location of the distribution density region considering road network density varies greatly. The statistics are shown in Table 6. Daoli and Xiangfang are more densely affected by the road network density, and Songbei has not changed substantially.

**Table 6.** Density region statistics considering the road network density.

|  | **Low to Medium** | **Medium to High** | **High** |
|---|---|---|---|
| Administrative District | Daoli, Xiangfang, Pingfang, Songbei | Xiangfang, Pingfang, Songbei | Pingfang |
| Area(km$^2$) | 21.1 | 5.2 | 4.6 |

### 5.1.3. Severity Distribution

The traffic safety level of urban roads is also related to casualties and property damage. Using the accident severity as the weight of each collision point, the weighted density of all points is calculated by Equation (8). By dividing weighted density by the density without weight, we obtain the severity distribution on a unit area. The results are shown in Figure 7.

Compared with the density distribution considering the road network density above, the spatial distribution has a substantial difference. Table 7 shows the statistical results. In Figure 6, the high-density area is located in the urban center of Pingfang. However, it is located in Daowai, Xiangfang, and Nangang in Figure 7. The medium-high density areas occupy a considerably large area. Most of the areas are located in the urban center of the district. The medium- and low-density areas have a wider distribution, covering an area of 136.2 km$^2$.

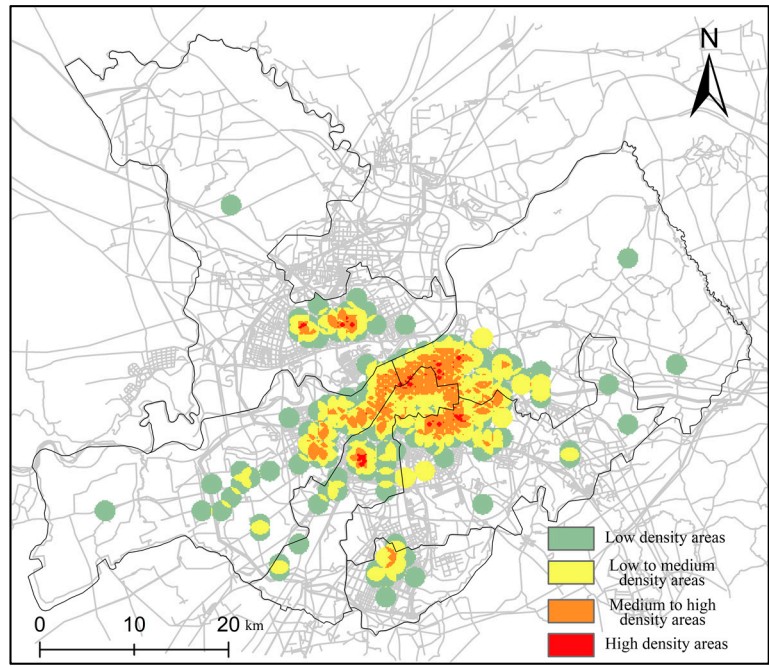

**Figure 7.** The density distribution of collision severity.

**Table 7.** Density region statistics of the severity.

|  | Low to Medium | Medium to High | High |
|---|---|---|---|
| Administrative District | Daoli, Daowai, Xiangfang, Nangang, Songbei, Pingfang | Daoli, Daowai, Xiangfang, Nangang, Songbei, Pingfang | Daowai, Nangang, Songbei, Pingfang |
| Area(km$^2$) | 136.2 | 78.7 | 4.1 |

　　Clustering analysis of the data obtained clustering spatial distribution of collisions based on the different levels of severity. The interpretation of the different clustering results is list as follows:

(1)　High severity clustering (high–high), which means that the spatial area contains many high severity collision points.

(2)　High–low severity clustering (high–low), which means that the spatial range of many low-severity collision points contains a few high severity collision points.

(3)　Low–high severity clustering (low–high), which means that the spatial range of many high-severity collision points contains a few low severity collision points.

(4)　Low severity clustering (low–low), which indicates that the spatial range contains many low-severity collision points.

(5)　Not-significant implies that there is no significant clustering relationship between the points in the region.

　　Figure 8 shows the results. A high–high clustering trend is shown in the urban center of Daoli and Daowai. In Songbei and Pingfang, most of the collision points demonstrate a high-low clustering trend. This result indicates that these areas generally have a low severity of collisions. The points with low-low clustering of severity existed only in the suburbs of Daoli.

　　In all of the above, the collisions are concentrated in the urban center of each district. Both the density and severity of rear-end collisions in the suburbs are low.

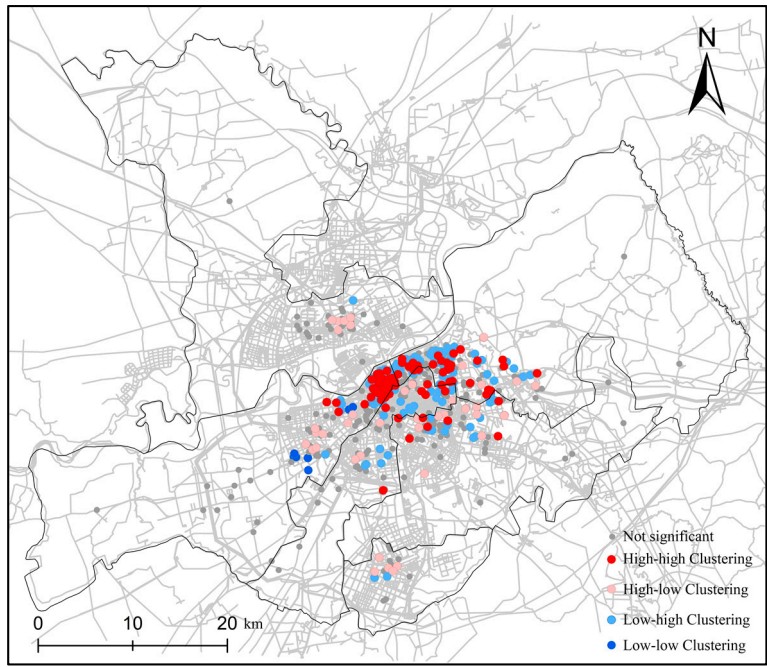

**Figure 8.** The clustering analysis of collision severity.

### 5.2. Time Distribution Characteristics

5.2.1. Seasonal Distribution

Harbin has distinct seasons. It has a long spring and winter and a short summer. Figure 9 shows the statistics of rear-end collisions from 2015 to 2019 according to the month. The high incidence month is in July, August and September. With August as the boundary, the number of collisions from February to August has an increasing trend. August to December has a decreasing trend. The maximum is 141, which occurs in August. The minimum is 55 in January.

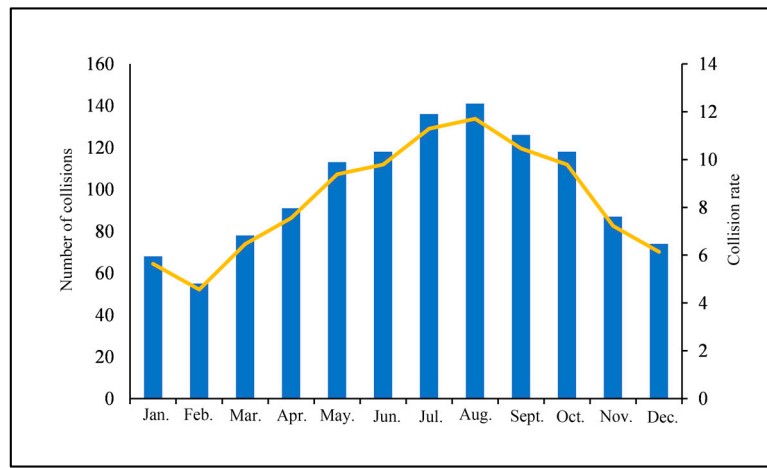

**Figure 9.** Monthly distribution.

Based on the geographical conditions and climate of Harbin, the season division is as follows: spring: March to May; summer: June to August; autumn: September to October; winter: November to February. By Equations (6) and (7), the density distribution in each season is calculated with or without considering the road network density. Figures 10 and 11 show the results in each season. The density distribution is concentrated in Daoli, Daowai and Xiangfang even in different seasons. These districts are typically prone to rear-end collisions.

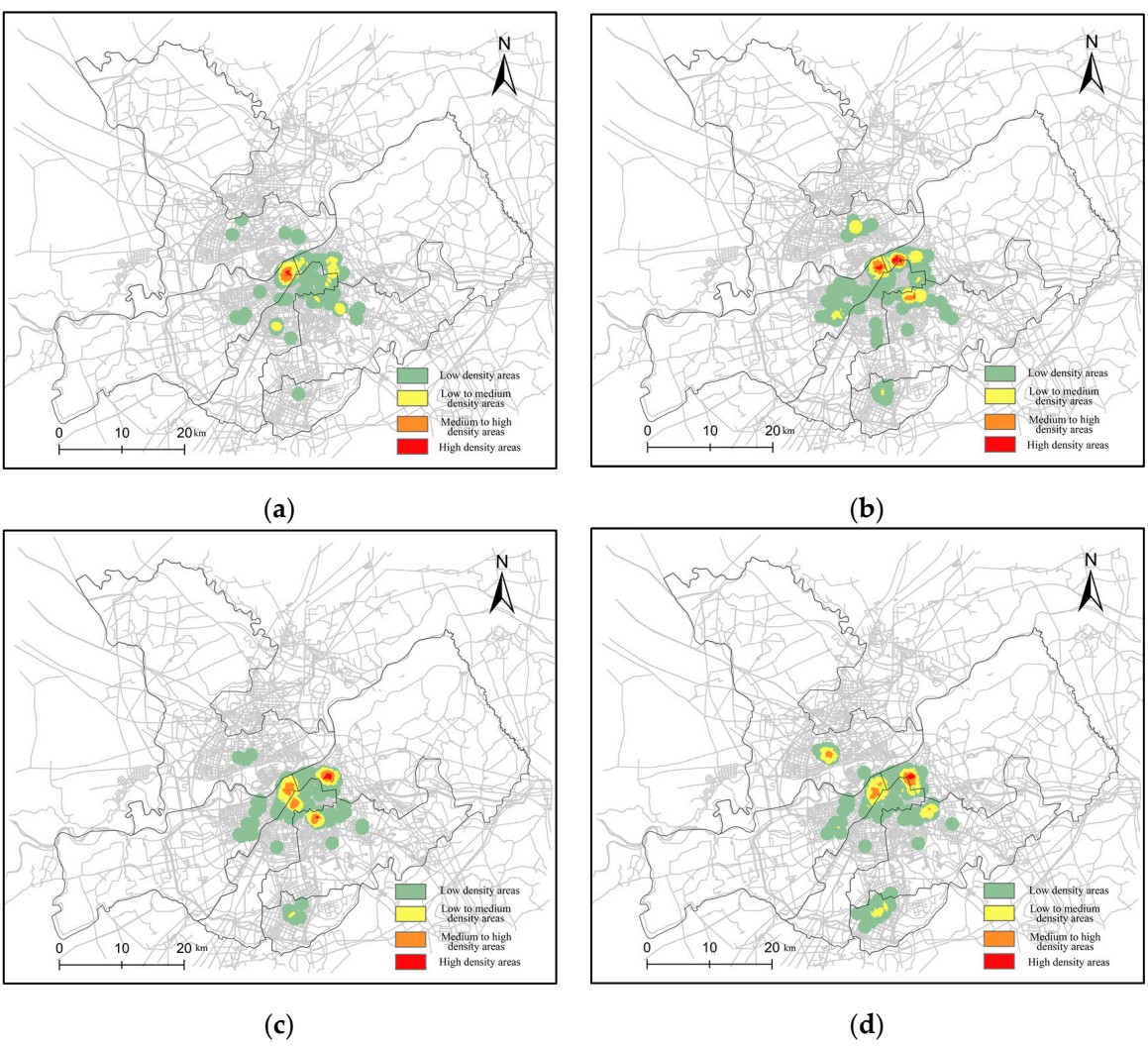

**Figure 10.** The density distribution in each season: (**a**) spring; (**b**) summer; (**c**) autumn; (**d**) winter.

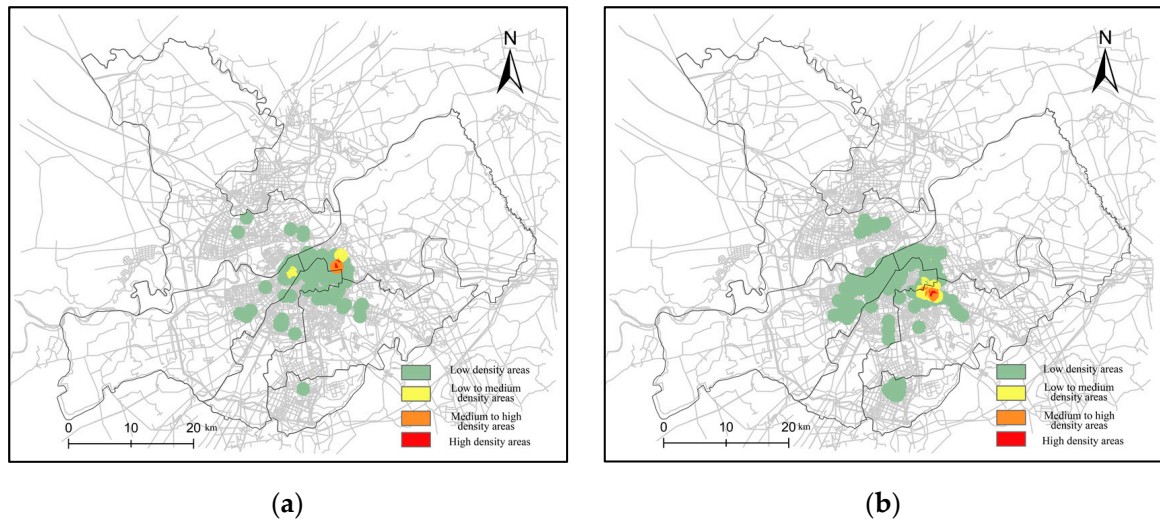

**Figure 11.** *Cont*.

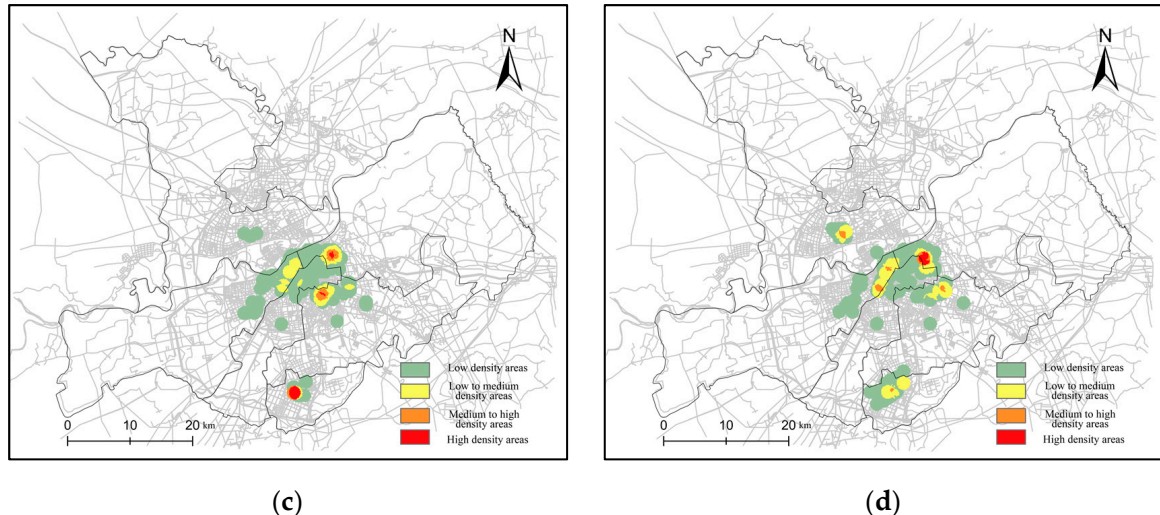

(**c**)              (**d**)

**Figure 11.** The density distribution in each season considering the road network density: (**a**) spring; (**b**) summer; (**c**) autumn; (**d**) winter.

### 5.2.2. Time Period Distribution

Figure 12 reports the time period distribution. The high incidence is concentrated from 6:00 to 10:00 and 16:00 to 20:00. The number is relatively small between 0:00 and 4:00. 10:00 and 16:00 are the boundaries. It has an increasing trend continuously before 10:00. The trend remains unchanged between 10:00 and 16:00. After 16:00, it decreases. The maximum is 159, which occurs from 8:00~10:00. The minimum is 27 from 2:00~4:00.

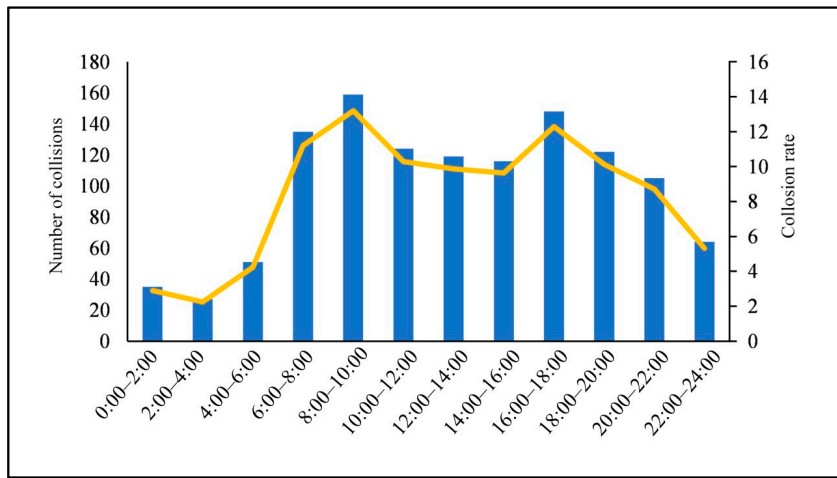

**Figure 12.** Time period distribution.

Based on the results above, the classification of time periods is 0:00~5:59, 6:00~11:59, 12:00~17:59 and 18:00~23:59. We can use Equations (6) and (7) to obtain the spatial distribution of each time period. Figure 13 shows the density distribution of each time period. Figure 14 shows the results considering the road network density. Daoli, Daowai and Xiangfang are still the high-density distribution areas. During commuting hours, Pingfang appears in high-density areas.

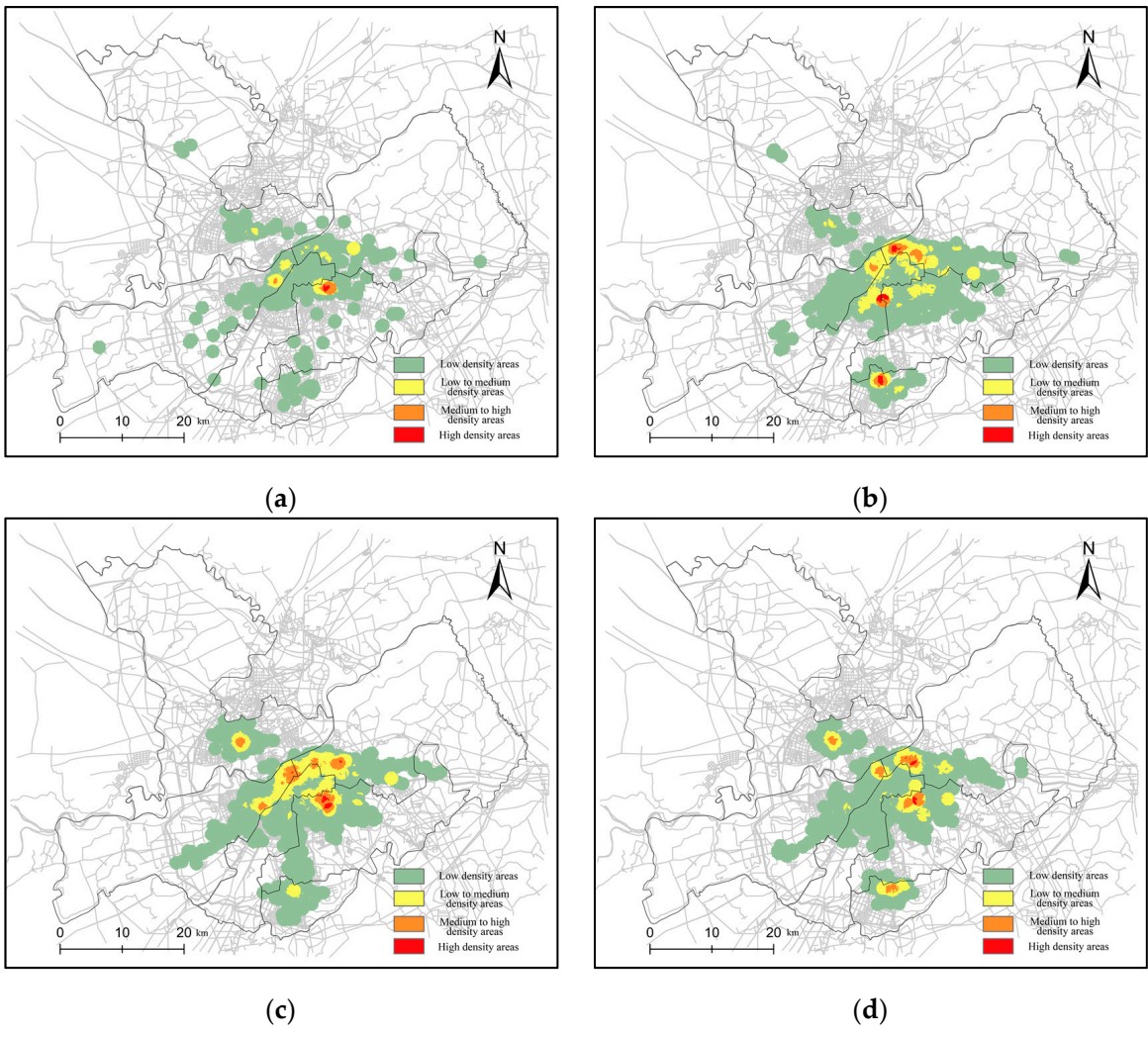

**Figure 13.** The density distribution in each time period: (**a**) 0:00~5:59; (**b**) 6:00~11:59; (**c**) 12:00~17:59; (**d**) 18:00~23:59.

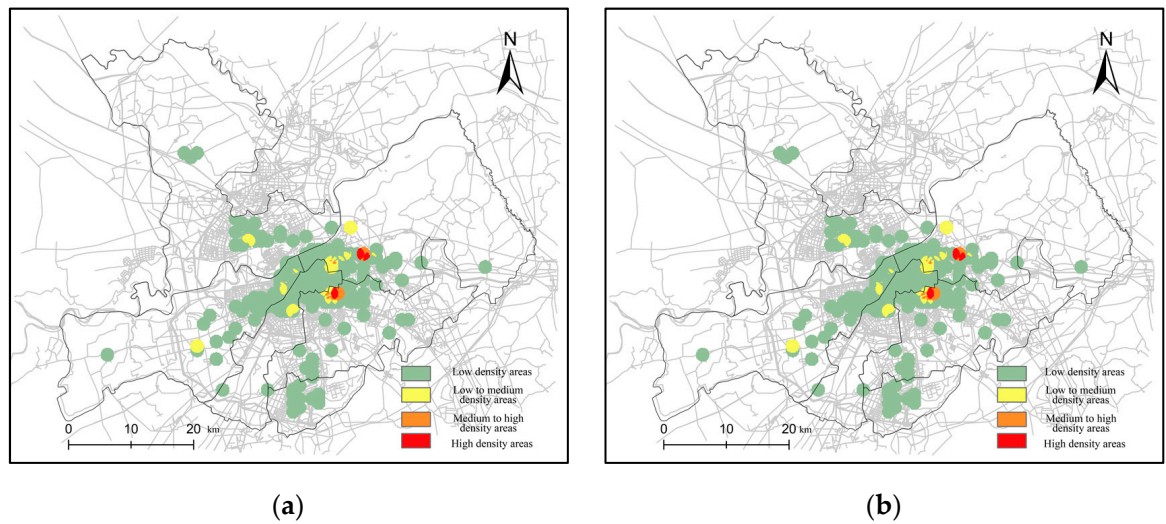

**Figure 14.** *Cont*.

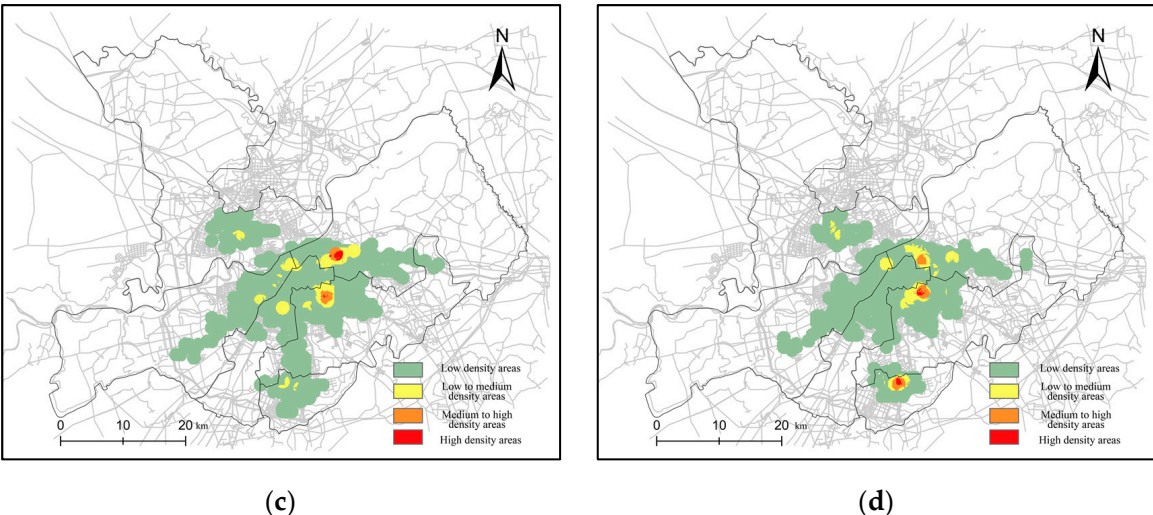

(**c**)                                          (**d**)

**Figure 14.** The density distribution in each time period considering the road network density: (**a**) 0:00~5:59; (**b**) 6:00~11:59; (**c**) 12:00~17:59; (**d**) 18:00~23:59.

## 6. Severity Prediction

### 6.1. LightGBM Prediction

The classification method follows traffic accident severity classification. We divided the test set as the proportion of 30% and the training set as 70%, 10 cross-validations are performed. The validation set is a uniformly random sample from the training set. This validation set was used for cross-validation. The parameter settings are shown in Table 8.

**Table 8.** LightGBM parameter settings.

| Parameter | Instruction | Value | Explanation |
|---|---|---|---|
| num_leaves | Number of leaf nodes | 40 | Number of leaf nodes is 40 |
| min_data_in_leaf | Data in each leaf node | 20 | 20 pieces data in each leaf node |
| objective | Task Type | multiclass | Multi-Classification |
| max_depth | Depth of decision tree | −1 | No restrictions |
| learning_rate | Learning Rate | 0.45 | Shrinkage rate is 0.45 |
| boosting | Boosting method | gbdt | Gradient Boosting Decision Tree |
| lambda_l1 | L1 regularization | 0.05 | L1 regularization weight is 0.05 |
| lambda_l2 | L2 regularization | 0.05 | L2 regularization weight is 0.05 |

First, we calibrated the number of LightGBM classifier features and the number of decision trees. Then, the accuracy and confusion matrix are selected to evaluate the classification accuracy of the LightGBM classifier. The results are shown in Figure 15 and Table 9.

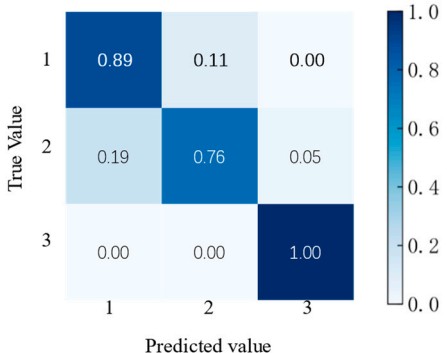

**Figure 15.** Confusion matrix accuracy of the LightGBM classifier.

**Table 9.** Confusion matrix of the LightGBM classifier.

| True Value | Classification Results | | |
| --- | --- | --- | --- |
| | **Property Damage Accident** | **Injury Accident** | **Fatal Accident** |
| Property damage accident | 173 | 22 | 0 |
| Injury accident | 25 | 100 | 6 |
| Fatal accident | 0 | 0 | 36 |

The accuracy rate of the LightGBM classifier is 85.36%, which can reach a high standard. The confusion matrix illustrates the following conclusions. For property damage accidents, the prediction accuracy rate is 89%. Eleven percent are incorrectly predicted as injury accidents. For injury accidents, the prediction accuracy rate is only 76%. A total of 19% are incorrectly predicted as property damage accidents, and 5% are predicted as fatal accidents. All fatal accidents in the test set are predicted correctly with an 100% accuracy rate.

Using the LightGBM classifier, the predicting characteristic curve is calculated. The AUC value under the curve is 0.89, as shown in Figure 16. When the AUC value is between 0.5 and 1, it indicates that the classifier is a more effective prediction.

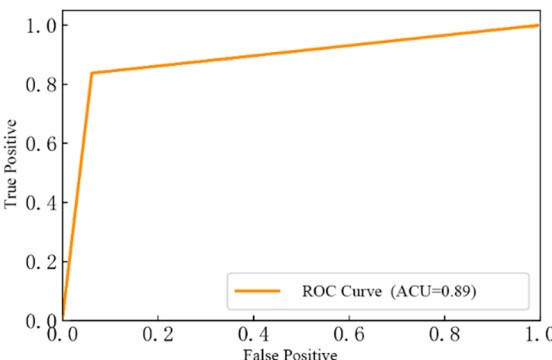

**Figure 16.** ROC curve of the LightGBM classifier.

Some of these predicted results are shown in Figure 17.

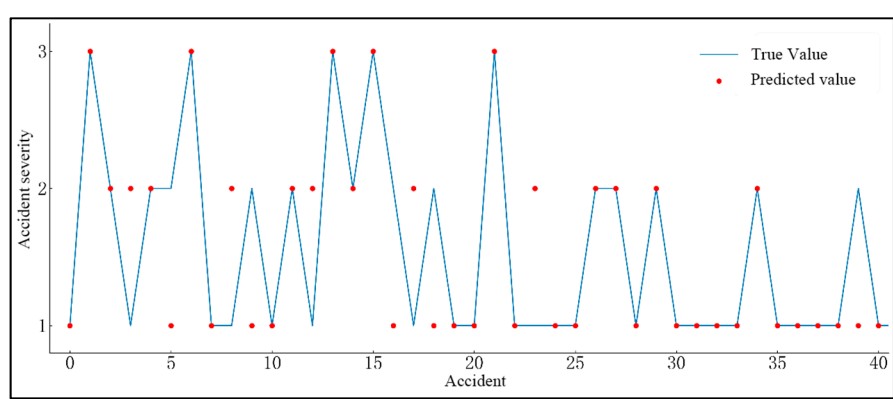

**Figure 17.** Prediction results of the LightGBM classifier.

### 6.2. Random Forest Prediction

The dataset contains nine attributing factors for each data item. This paper mainly uses the min–max normalization method (Equation (14)) to transform each factor value. After normalization, all values are in the range of [0, 1].

$$x^* = \frac{x - x_{min}}{x_{max} - x_{min}} \tag{14}$$

$x_{max}$ is the maximum value in the feature sample. $x_{min}$ is the minimum value. $x^*$ represents the new value after normalization.

The test set and training set were divided by proportions of 30% and 70%. As the same for the LightGBM classifier, RF classifier performed 10 cross-validations. The validation set was derived from the training set. RF parameter settings are shown in Table 10.

**Table 10.** RF parameter settings.

| Parameter | Instruction | Value | Explanation |
|---|---|---|---|
| n_estimator | Number of trees in RF | 10–2000 | 10–2000 trees by testing |
| max_features | Maximum number of features in partitioning | 9 | Considering all 9 features |
| criterion | Evaluation criteria for features in splitting | Gini | Gini coefficient |
| max_depth | Depth of decision tree | No input | No restrictions |
| min_samples_leaf | Minimum number of samples in leaf nodes | 1 | Less than 1 samples will be pruned |
| max_samples_split | Minimum number of samples for internal node repartitioning | 2 | Less than 2 samples will not be split again |
| max_leaf_nodes | Maximum number of leaf nodes | None | No restrictions |
| min_weight_fraction_leaf | Minimum sample weight of leaf nodes | 0 | No consideration |
| min_impurity_split | Minimum impurity of node segmentation | $1 \times 10^{-7}$ | Less than $1 \times 10^{-7}$ will not generate child nodes |

Because each experiment has result errors, multiple experiments were used to take the average. Each set of data derived from the experiment is the mean value of 10 experiments.

The number of trees in the RF classifier is an important parameter to be adjusted. To avoid over-fitting, in the random forest model, we selected a series number of trees to test the hyperparameters. Figure 18 shows the changes.

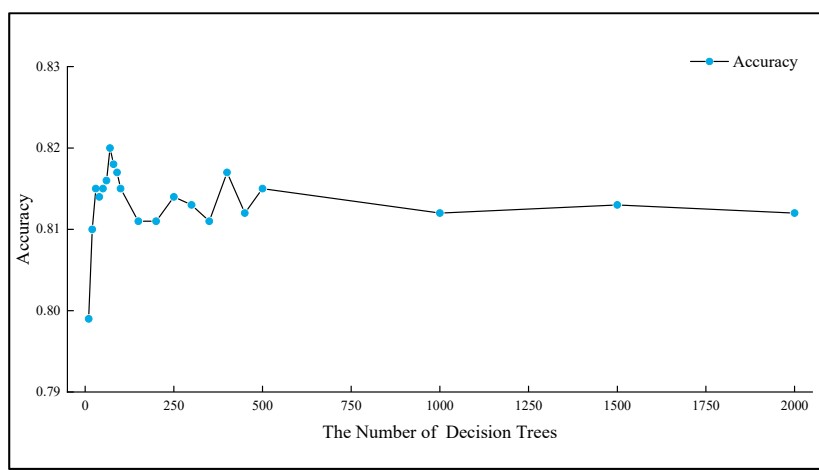

**Figure 18.** Accuracy of the RF classifier experiment.

In Figure 18, according to the increasing number of decision trees, the prediction accuracy tends to be smoother. The accuracy fluctuates between 81.1% and 81.5%. When more than 500 trees are used, the accuracy stabilizes at 81.4%. Therefore, we selected 1000 trees. After calibrating the number of RF classifier features and decision trees, the accuracy and confusion matrix can be obtained. The results are shown in Table 11 and Figure 19.

**Table 11.** Confusion matrix of the RF classifier.

| True Value | Classification Results | | |
| --- | --- | --- | --- |
| | **Property Damage Accident** | **Injury Accident** | **Fatal Accident** |
| Property damage accident | 169 | 14 | 2 |
| Injury accident | 51 | 83 | 1 |
| Fatal accident | 1 | 0 | 41 |

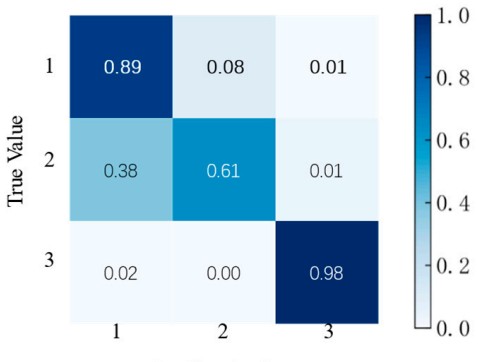

**Figure 19.** Confusion matrix accuracy of the RF classifier.

The accuracy rate of the RF classifier is 81.43%. For property damage accidents, the prediction accuracy rate is 91%. Only 8% are incorrectly predicted as injury accidents, and 1% are predicted as fatal accidents. However, the prediction accuracy rate for injury accidents is only 62%. Thirty-eight percent are incorrectly predicted as property damage accidents, and 1% is determined to be fatal accidents. For fatal accidents, the accuracy rate is 98%, and 2% are incorrectly predicted as property damage accidents.

In Figure 20, the AUC value is 0.83. This result shows that the classifier can predict well.

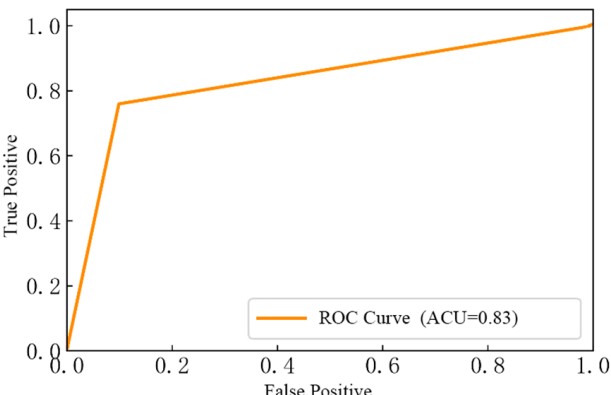

**Figure 20.** ROC curve of the RF classifier.

Some of these predicted results are shown in Figure 21.

### 6.3. Analysis of Rear-End Collision Causes

Based on the above, the LightGBM classifier has higher prediction accuracy. We selected the LightGBM classifier to rank the nine parameters in degree of importance. Table 12 shows the results.

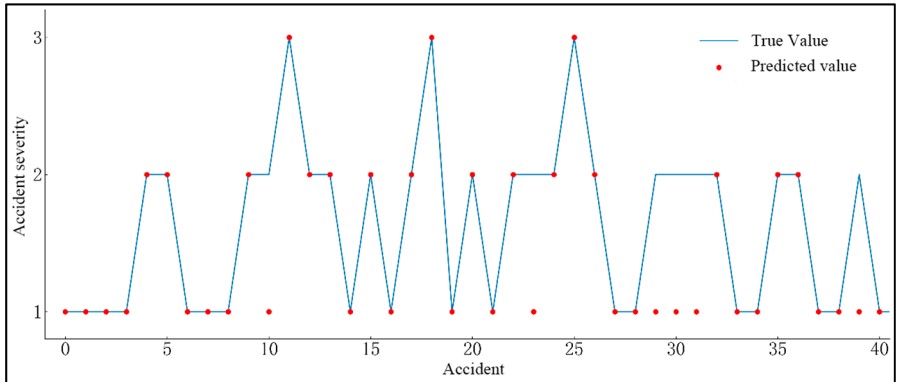

**Figure 21.** Prediction results of the RF classifier.

**Table 12.** Ranking of feature importance.

| Ranking | Characteristic Parameters | Importance Level |
|---------|---------------------------|------------------|
| 1 | Temperature | 0.5643 |
| 2 | Weather | 0.1036 |
| 3 | Time | 0.0867 |
| 4 | Vehicle Type | 0.0698 |
| 5 | Wind Speed | 0.0654 |
| 6 | Season | 0.0554 |
| 7 | Location | 0.0329 |
| 8 | Week | 0.0214 |
| 9 | Accident Type | 0.0003 |

The most important parameter affecting the severity of rear-end collisions is temperature, followed by weather, time, vehicle type, wind speed, season, location, week, and accident type. Therefore, traffic safety managers need to especially focus on environmental factors, such as temperature and weather. It is the key to reduce the probability of rear-end collisions in Harbin.

## 7. Discussion

The results show that the occurrence of rear-end collisions are strongly characterized by spatial and temporal distributions. In summary, the main points are as follows:

(1) It has a high frequency in the urban center of districts.
(2) In the morning, collisions have a high probability.
(3) It shows a significant correlation between spatial features and rear-end collisions severity.
(4) The probability of collisions is higher on main roads with district intersections.

In fact, the urban center of Daoli and Daowai is the old urban area of Harbin. Many old roads and one-way roads lead to complicated driving conditions. There are more serious traffic problems in this area. Pingfang shows high density only in commuting time. Working commuting areas in Pingfang are more concentrated than in other districts. The conclusions drawn from the above analysis are the same.

Compared to the existing studies, there are several similar conclusions that can be drawn. Dimitrioua et al. [41] illustrate rear-end collisions potential was presented when traffic flow and speed standard deviation were higher. Large traffic flow and vehicle speed differences often occur in urban centers and during daytime working hours. Point (1) and (2) confirms the conclusion. Point (3) is similar to the view of Liu and Sharma [12]. The spatial characteristics of the city can be considered as a factor in the occurrence of rear-end collisions. Soltani and Askari [3] obtained the following conclusions. The urban areas that connect to each other areas were determined as clusters with high crash rates. Recreational facilities and schools along the road are associated with the occurrence of traffic accidents. This is similar to the Point (4). In our results, the administrative border area between Daoli

and Daowai is the hot spot. It is reasonable to draw such a result considering that Daoli and Dawai, as old urban areas, have a high density of buildings and urban facilities.

Comparing the LightGBM and RF classifiers are can be seen in Table 13. The LightGBM classifier has better prediction accuracy for overall accuracy. The RF classifier has better prediction accuracy for property damage accident prediction accuracy.

**Table 13.** Comparison of prediction accuracy.

|  | Prediction Accuracy | | | |
|---|---|---|---|---|
|  | **Overall** | **Property Damage Accident** | **Injury Accident** | **Fatal Accident** |
| LightGBM | 0.85 | 0.89 | 0.76 | 1 |
| RF | 0.81 | 0.91 | 0.61 | 0.98 |

This paper concludes that temperature shows a substantial effect on the cause of rear-end collisions. The temperature characteristics of Harbin are very important. In autumn, there is a large difference in temperature between day and night. The winter is quite long. There is large amounts of snow and ice on roads. The concentration of rear-end collisions in autumn is linked to the change in temperature. In addition to driving factors, temperature has a considerable influence on the severity of collisions at road intersections [6]. In comparison, rear-end collisions cause factors that exclude season and adverse weather in North Carolina [42]. North Carolina has a relatively mild climate temperature difference, which is markedly different from Harbin.

## 8. Conclusions

This paper used spatiotemporal analysis and machine learning to analyze rear-end collision data. The results highlight some characteristics of rear-end collisions.

Spatial distribution is characterized by three aspects: spatial distribution, density distribution and severity distribution. The spatial distribution is concentrated in administrative border areas between Daoli and Daowai. Considering the density distribution only, collisions are concentrated in Daoli and Xiangfang. Considering the effect of road network density, collisions are concentrated in Pingfang. An analysis of the severity and clustering of rear-end collisions shows a high–high clustering trend in the urban centers of Daoli and Daowai. In summary, administrative intersection areas with dense traffic are high incidence areas.

The temporal distribution characteristics were analyzed by seasonality and time period. Rear-end collisions mostly occur from July to September. In autumn, rear-end collisions are distributed over the largest range and at high density. In winter, all administrative districts have a high density area. The results of the time period characteristics are more obvious. Regardless of whether road network density is considered, the occurrence of rear-end collisions are concentrated between 6:00~11:59.

We then made predictions by the LightGBM and RF classifiers. Comparing the accuracy of the two classifiers, LightGBM has the highest combined accuracy. We calculated the feature importance ranking of affecting accident severity with the LightGBM classifier. It was concluded that temperature is the most important factor.

In this paper, the data have some limitations in describing the difference in collisions. Data descriptions are only available for the external environment and lack specific details. Therefore, the analysis of the causal factor is not comprehensive enough.

Further studies will gather more details about the collision, such as vehicle conditions, traffic conditions and road types. The LightGBM and RF classifiers belong to machine learning based on the decision tree. The machine learning prediction model can be considered combined with a clustering algorithm to mine the potential causes. This method can be applied to the analysis of traffic accidents in many scenarios, such as traffic accidents at urban road intersections, crashes of motor vehicle and non-vehicle and freeway rear-end

accidents. These all can be analyzed using this method for spatiotemporal characteristics and accident causes.

**Author Contributions:** Conceptualization, W.Z.; methodology, W.Z. and T.L.; software, T.L. and J.Y.; validation, W.Z., T.L. and J.Y.; formal analysis, T.L.; investigation, W.Z. and T.L.; resources, W.Z.; data curation, T.L.; writing—original draft preparation, W.Z and T.L.; writing—review and editing, T.L.; visualization, T.L. and J.Y.; supervision, W.Z.; project administration, W.Z.; funding acquisition, W.Z. All authors have read and agreed to the published version of the manuscript.

**Funding:** This research was funded by the Fundamental Research Funds for the Central Universities Category D Project Carbon Neutralization Project, grant number 2572021DT09.

**Institutional Review Board Statement:** Ethical approval was waived as the experiment would not cause any mental injury to the participants, have any negative social impact, or affect the participants' subsequent behaviors. Although our research institutions do not have an appropriate ethics review board, several experts have discussed the research plan as sound and feasible.

**Informed Consent Statement:** Informed consent was obtained from all subjects involved in the study.

**Data Availability Statement:** The data used to support the findings of this study are available from the corresponding author upon request.

**Conflicts of Interest:** The authors declare no conflict of interest.

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
