# Peer review of "Exploring the Spatiotemporal Characteristics and Causes of Rear-End Collisions on Urban Roadways"

_sustainability, doi:10.3390/su141811761_

Round 1

Reviewer 1 Report

It is an interesting and mostly well written paper on the statistical characteristics of rear-end collision accidents. However, some things are less clear in the method description and result presentation, and needs clarification in the paper.

I have three major concerns about the paper:

Section 3 Methods: In general, the method description is unbalanced - detailed equations for calculating center of mass (which is simple and probably well known) and very brief and without formal equations for the LightGBM (which is a much more complicated method and probably less known among readers). It makes it hard to follow the method descriptions unless the methods are known in advance.  Specific recommendation: Also for the simple ones, please give the user the context by specifying for what it will be needed later. Eg, I can see from the SDE equations that they assume bi-variate data, whereas in general you might want to calculate a multi-dimensional ellipsoid for high dimensional data. It will make sense if you specify here that you will only use it for spatial coordinates, not all of you data. For the Morans I, you just use the concept of High-high, High-low, etc, clustering without explaining what it is and what it conceptually means. Please add that description, since all readers are not familiar with this. For the LightBGM the description is very abstract and refers to a lot of other methods. "another evolutionary form of Gradient Boosting Decision Tree" - does not make sense since you have not described GBDT. You must expand this whole description to at least make it intuitively clear what the model does. (Adding equations will not help here, text is better) For the Random Forest it is also abstract, and then you suddenly throw in some equations for the Bootstrap resampling, which makes no sense when the rest is so cursorily described. Extend this too in an intuitive way, and explain where Bootstrap is needed.

I do not understand how you get the Pearson correlation matrix on Page 8. First, you can not use a single coefficient for multi-valued discrete variables. It does not make sense to code it as a continuous value if the values are not expected to be ordered. Seasons are cyclic. Vehicle types can not easily be ordered. Further, what strange rule for interpretation of the coefficient do you use: More than 0.8 is a linear correlation, but less than 0.8 "has no effect". I would say that anything with absolute value above 0.5 is quite strong, such as between Season and Temperature (although again, Season can not be linearly ordered the way you do - it would maybe make more sense to make summer=1, Winter=3, and Fall and Spring both=2). And maybe most important, how is this matrix used? Just to remove superfluous features? The features are few and the models used should be sufficiently good at removing redundant features themselves.

My strongest concern though is that the results are suspiciously good. Do you really claim that it is sufficient to just look at your input data, such as Season, Weather, Time of day, etc, and then tell almost certainly whether the incident was fatal or not? Is this reasonable? Have you really performed validation in the proper way, or is this a typical case of overfitting? You present the sizes of training set and test set. Were there no separate validation set? Were cross validation used? How exactly were the training and test set used during the experiments? Is it absolutely certain that the Test set was not used to fine-tune or select hyperparameters, or as stopping criterion? Can you please report both the results on the Training set and the Test set. 

Preferably, to make sure there is no over-fitting going on, you should keep away a part of the data as a validation set, which is never used during construction of the models. Then you use the test set for adjusting model parameters and possibly stopping condition. When you are happy with the model, then is the first time you open your validation set to evaluate the performance on it. 

Some minor additional comments:

Page 3, line 101: What do you mean by "combining neural networks and machine learning"? Neural networks is one specific type of machine learning.

Line 107: "a comparison between the two machine learning models". You have not yet specified which two models, so I suggest removing "the", and inserting the names of the models, so it reads "a comparison between two machine learning models, LightGBM and Random Forest, ...".

Page 5, table 1: "Basic info" is listed twice (at the beginning and at the end).

Page 8, figure 4: Why is the standard deviation ellipse not centered at the average center? And what do you mean that the short axis shows that it is more clustered? The short and long axis may be more indicative of the shape of the studied region than of any "cluster property" of the actual accidents.

Page 11: Again, you need to explain the significance of the High-high, High-low etc clustering. It is hard to understand these results otherwise.

Page 13: Section 5.2.2 should be called "Time period distribution"

Page 18: Last paragraph, missing "T" in "The" at the beginning of the sentence. 

Page 19: End of second paragraph, repeats that LightGBM is best, which is already mentioned two sentences back. Just remove the last sentence.

Page 19, line 405: A period is on the wrong side of the space, " .Comparing".

Reviewer 2 Report

The topic of the paper is very interesting and useful. The paper is very nicely written. However, I suggest that the paper be further improved before publication in the journal.

 ·   The literature is nicely systematized; however, the influence of the human factor is missing. I suggest you cite papers that report driver speed/hazard perception as a contributor to rear-end crashes.

Li, Y., Zheng, Y., Wang, J., Kodaka, K., & Li, K. (2018). Crash probability estimation via quantifying driver hazard perception. Accident Analysis & Prevention, 116, 116-125.

Čubranić-Dobrodolac, M., Švadlenka, L., Čičević, S., Trifunović, A., & Dobrodolac, M. (2022). A bee colony optimization (BCO) and type-2 fuzzy approach to measuring the impact of speed perception on motor vehicle crash involvement. Soft Computing, 26(9), 4463-4486.

Mahajan, K., & Velaga, N. R. (2021). Sleep-deprived car-following: indicators of rear-end crash potential. Accident Analysis & Prevention, 156, 106123.

· In data processing, you used 1266 traffic accidents as a sample. Before that, you wrote that the database consisted of 1372 cases. It is necessary to clearly state in one sentence the criteria by which you removed part of the traffic accidents from the sample. So that the readers would not be confused.

·  I appreciate the way the results are presented (tables, graphs, figures…).

·  Please expand the discussion section by one paragraph to compare your results with other studies. I understand that there are not many studies on this topic, but maybe the papers I suggested can help you.

·  I appreciate the limitations of the study and directions for future research.

· In the conclusion, you stated, "This method can also be applied to other types of traffic accidents". Expand this section to include other traffic accidents (e.g., side collisions, other categories of road users, etc.). Where else can you implement the results?

Round 2

Reviewer 1 Report

The authors have carefully considered most of my concerns. The paper is now in a much better shape. I have just a few remaining comments, that I would like to see fixed:

Section 3.3, Lines 172 - 174: With the reservation that this High/Low concept was new to me so I may be mistaken: Are you sure that it is not High-high and Low-low which indicate high correlation with neighbors (either all points are high or all points are low), whereas High-low and Low-high are less correlated (some are high and some are low in the neighborhood)?

Section 4.2, line 223 vs line 231 vs last line of table 4: There is some confusion about the actual number of data samples used in the end. First it says 1236. Then it says that 1127 samples were retained after coordinate transformation. Is this only because the others fell outside the area in the figure, but they are still in the analysis? Or were they discarded from analysis as well? And then on the last line in the table (Temperature) it says that 100% of the data was 1205 samples. So how many was it really?

Figure 4: The ellipse is still not centered around the mean of accident positions. Since the ellipse it calculated as the spread around the mean value (see line 135 in your paper where "a" and "b" are defined relative the center), it should also be visualized centered at this position. You can almost see in figure 4 that you have plenty of accidents below the ellipse but none above. Centering it at the mean will make it cover the accidents better, with a balance of a few above and a few below.

Section 6: Thank you for clarifying the used protocol with training and cross validation to me. You could well have described what you explained to me also in the paper (optional though). It still would have been nice to see the results both on the test set and the training set (also optional).

However, there is a confusing claim in line 380:
"All fatal accidents in the *training set* are predicted correctly"
Regardless of whether you add more details on the testing protocol, you must make very clear early in this passage that the results are on the *test* set. This line is probably a typo. If not, you *must* present separately the results on both the test and training sets.

(Table 10: By trimming the texts and playing with line breaks you can probably make the table less wide in the page. Just to make it look better.)

Section 7, line 442: "It shows a have significant", remove the superfluous "have".

Round 3

Reviewer 1 Report

Thanks for the explanations and clarifications. I have no more comments.